# UrbanDataLayer: A Unified Data Pipeline for Urban Science

**Yiheng Wang** [1], **Tianyu Wang** [1], **Yuying Zhang** [1]
**Hongji Zhang** [1], **Haoyu Zheng** [1], **Guanjie Zheng** [*,1], **Linghe Kong** [*,1]
[1] Shanghai Jiao Tong University, Shanghai, China
{yhwang0828, wty500, shjtzyy01, zhanghongji,
langanzheng, gjzheng, linghe.kong}@sjtu.edu.cn

## Abstract

The rapid progression of urbanization has generated a diverse array of urban data, facilitating significant advancements in urban science and urban computing. Current studies often work on separate problems case by case using diverse data, e.g., air quality prediction, and built-up areas classification. This fragmented approach hinders the urban research field from advancing at the pace observed in Computer Vision and Natural Language Processing, due to two primary reasons. On the one hand, the diverse data processing steps lead to the lack of large-scale benchmarks and therefore decelerate iterative methodology improvement on a single problem. On the other hand, the disparity in multi-modal data formats hinders the combination of the related modal data to stimulate more research findings. To address these challenges, we propose UrbanDataLayer (UDL), a suite of standardized data structures and pipelines for city data engineering, providing a unified data format for researchers. This allows researchers to easily build up large-scale benchmarks and combine multi-modal data, thus expediting the development of multi-modal urban foundation models. To verify the effectiveness of our work, we present four distinct urban problem tasks utilizing the proposed data layer. UrbanDataLayer aims to enhance standardization and operational efficiency within the urban science research community. The examples and source code are available at `https://github.com/SJTU-CILAB/udl`.

## 1 Introduction

The accelerated pace of urbanization has enhanced life quality while concurrently inducing issues such as air pollution and traffic congestion. Extensive urban data has been recorded due to the widespread use of advanced sensing technologies [29]. Simultaneously, urban studies have sprung up among various domains of human mobility [15], air quality [6, 20], traffic dynamics [18], climate change [35], spatial planning [48] and poverty [42, 32], etc. However, several *challenges* are posed.

*Firstly, numerous urban studies work on separate problems using different datasets case by case or performing different processings on the same dataset. This lack of standard benchmarks hinders the overall improvement of research.* In urban issues, researchers often self-define the problem and propose methods accordingly. Based on an analysis of 88 papers published in seven AI conferences shown in Fig. 1, three phenomena are observed. (1) Many urban problems are defined within the same domain, yet disparate datasets are used for identical problems. (2) Even if they use the same datasets,

---

[*]Corresponding Author.

Submitted to the 38th Conference on Neural Information Processing Systems (NeurIPS 2024) Track on Datasets and Benchmarks. Do not distribute.

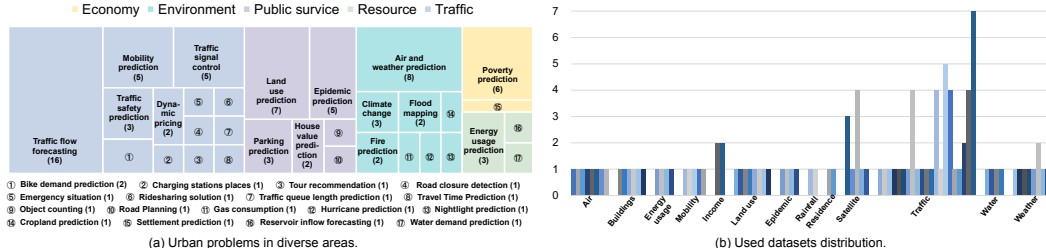

Figure 1: Problems and datasets in published papers. (a) Urban problems studied in five areas: traffic, public service, environment, ecomomy and resource (from left to right). The numbers below are the count of relevant papers. (b) Dataset types for each category. Each bar represents the number of papers using that dataset. Papers use data of different datasets in similar domains and the distribution of datasets is very decentralized.

variations in data processing might lead to inconsistent experimental data. (3) More differences in the final experimental data may also exist that are not known due to the data not being publicly available. Even in relatively mature urban tasks such as urban spatial-temporal prediction, only less than 30% of the papers have made their data public [36]. As shown in Table 1, a significant portion of experimental data in urban studies remains inaccessible. This phenomenon makes comparisons between these methods difficult and the results are hard to reproduce due to non-public experimental data. Furthermore, researchers cannot continuously improve the performance of the methods under the same standard, which hinders the progress of urban research.

*Secondly, urban data exists in multiple modalities, miscellaneous formats, and non-uniform granularity, and involves cumbersome processing; urban research often requires multiple data fusions. Repetitive and intricate data processing is troublesome and prone to errors, making data utilization poor.* Fusing knowledge from different datasets is effective and essential in urban research. Unlike Computer Vision and Natural Language Processing which have standardized datasets such as ImageNet [7] and WikiText-103 [30], urban datasets frequently adopt distinct storage formats with diverse granularities, encompassing images, tables, trajectories, points, and beyond. This challenge impedes researchers especially novices in the domain of efficiently and correctly combining and leveraging the data, which introduces obstacles in large-scale urban research.

Therefore, we propose an effective and efficient urban data management suite named UrbanDataLayer (UDL), which provides five standard urban data layers and efficient data processing tools with the following characteristics. (1) **Reproducible benchmark:** People can utilize UDL to easily process their data, make it a public benchmark, and compare with SOTA methods. (2) **Combinable multi-modal data:** We provide examples of combining urban data with spatio-temporal base data, e.g., satellite image and road network data to create the possibility for multi-modal spatio-temporal foundation model building. (3) **Extensibility:** UDL can be expanded in both spatio-temporal and feature dimensions and encourages researchers to fill in the gaps of absent universal urban data.

## 2 Related Work

In contrast to other domains like Computer Vision, Natural Language Processing or tasks like Graph Node Classification have common datasets such as ImageNet [7] and CIFAR-10 [17], WikiText-103 [30], Cora [27], respectively. Regrettably, urban computing research lacks common datasets and data formats and somewhat inhibits the advancement of this field.

It has recently come to our attention that there is a benchmark LibCity [37] for solving urban spatio-temporal prediction problems. It includes pivotal stages related to traffic prediction into a systematic pipeline and provides 40 diverse datasets of unified storge format. It merely focuses on scenarios of urban traffic and does not cover all types of data in urban.

Data produced within urban areas typically exhibits an association with either spatial or spatiotemporal attributes [47]. Datasets originating from diverse domains present different structures, resulting in

Table 1: Data used in published research. The research of the same field works on separate datasets and most of them are not public. Take economy, air, and traffic domain as examples.

| Domain | Data | Time span | Spatial coverage | Paper | Type | Used Time | Used Space | Public* |
|---|---|---|---|---|---|---|---|---|
| Economy | Digital Globe Worldview Satellite | - | Global | [13] | Polygon | - | South Korea | ✗ |
| | Villages images from Google Maps | 2011 | Global | [32] | Grid | 2011 | India | ✗ |
| | Nightlight from NOAA | 2013 | Global | [42] | Grid | 2013 | Africa | ✗ |
| | Nightlight from NASA | 2012 | Global | [28] | Grid | 2012 | Global | ✔ |
| | Expenditure (poverty) from LSMS | 2011 - 2012 | Uganda | [42] | Grid | 2011 - 2012 | Uganda | ✗ |
| | | | | [2] | Grid | 2011 - 2012 | Uganda | ✗ |
| | Urban LIA (low-income areas) Data | - | Kisumu, Malindi, Nakuru | [19] | Point | - | Kisumu, Malindi, Nakuru | ✗ |
| | Income statistics from SECC | 2011 | India | [32] | Grid | 2011 | India | ✗ |
| Air | KDD CUP of Fresh Air | Jan. 1, 2017 - Apr. 30, 2018 | Beijing | [12] | Graph | Jan. 1, 2017 - Apr. 30, 2018 | Beijing | ✗ |
| | Urban Air data | Aug. 2012 - May. 2015 | 302 Chinese cities | [49] | Point | Aug. 2012 - May. 2015 | Chinese mainland | ✗ |
| | | | | [6] | Point | May. 1, 2014 - Apr. 30, 2015 | Beijing | ✗ |
| | | Jan. 1, 2015 - Dec. 31, 2018 | Chinese mainland | [20] | Point | Jan. 1, 2015 - Dec. 31, 2018 | Chinese mainland | ✔ |
| Traffic | NYC-Taxi | Jan. 1, 2015 - Mar. 1, 2015 | New York City | [43] | Grid | Jan. 1, 2015 - Mar. 1, 2015 | New York City | ✔ |
| | | | | [45] | Grid | Jan. 1, 2015 - Mar. 1, 2015 | New York City | ✗ |
| | Traffic dataset from Caltrans | 2015 - 2016 | San Francisco | [41] | Graph | 2015 - 2016 | San Francisco | ✗ |

*Whether the processed data in the paper is public.

different representations. When confronting a problem, it is customary to extract knowledge from numerous diverse datasets by data fusion. In particular, the recently proposed time-series large models [40, 11] frequently fuse data from different domains to obtain knowledge.

In the last decades, work like Open Geospatial Consortium [1] has been dedicated to establishing standards for geospatial data which is also related to urban data. However, the standards assembled as OGC APIs are designed primarily for geospatial data's release and access, which can be viewed more as a kind of "raw data". Unlike OGC, UDL aims to define an urban data pipeline that can process and fuse data as input directly into the model. In addition, it is not limited to geospatial data and other urban data like time series data are also in this scope.

## 3 UrbanDataLayer: A Data Suite for Urban Research

### 3.1 UDL layer-wise pipeline

The UDL (UrbanDataLayer) is a suite of standard data structures and pipelines for city data engineering, which processes city data from various raw data into a unified data format. The datasets used in one research may have different types and formats, and often come from different sources [23]. Urban data inputs into the UDL undergo a series of transformations, including conversion from raw data to standardized data layers, re-alignment of granularity, and fusion of disparate datasets, before being utilized and stored. Consequently, we delineate **four stages of data wrapping** and **three data processing steps** within the UDL, as depicted in Fig. 2.

The four data wrappers represent four stages in the data processing pipeline, transitioning from raw data to fused data that can be directly utilized by models. These stages include the raw data source, standard data layer, granularity-aligned data, and fused data, respectively. In standard data layer which is the main component of UDL, we summarize the urban data into five data structures: grid, graph, point, linestring and polygon. The details of each data layer are provided in the documentation[1].

The UrbanDataLayer builds the data layers and user-friendly APIs, simplifying the processing and reuse of city data in urban research. As depicted in Fig. 2, the components of UrbanDataLayer between four data wrappers are scheme transformation, granularity alignment, and feature fusion.

---

[1]https://urbandatalayer-doc.readthedocs.io/en/latest/

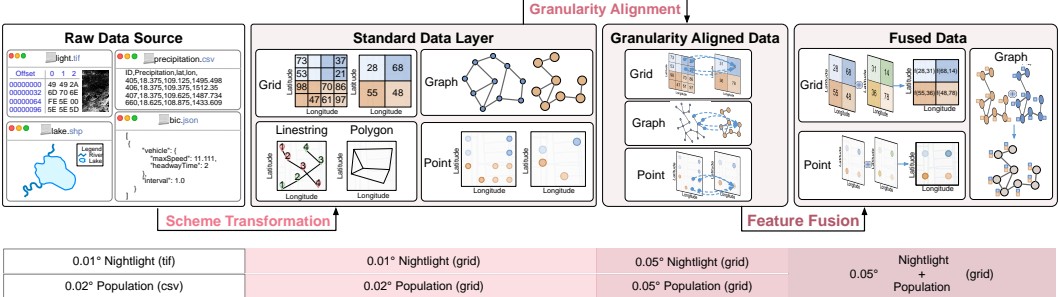

Figure 2: Overview of UrbanDataLayer framework. The words in red are the data processing steps.

In contemporary urban computing, datasets from diverse domains increasingly exhibit interconnections influenced by complex underlying relationships [46], underscoring the need for effective data fusion techniques to capture and leverage these connections. To exemplify the application of UDL (depicted at the bottom of Fig. 2), let's consider an example. Given nightlight data and population data in different formats and granularities, we aim to derive fused data for future downstream tasks. The process unfolds as follows: Firstly, we obtain standard grid data while preserving the original granularity through Scheme Transformation. Next, we acquire the target granularity data via Granularity Alignment. Subsequently, the fused data can be extracted through Feature Fusion. The entire process is managed by UDL.

## 3.2 General functionalities

For the defined five types of UDL layers, data operations like constructing, modifying and querying data by coordinates are provided. Besides this, users can easily access common data processing methods through UDL interfaces. The main types of interfaces are as follows: (1)*Scheme Transformation*: Facilitates the transfer of raw data to UDL data and between data layers (Fig. 3). (2)*Granularity Alignment*: Converts a standard data layer into different spatial granularities. (3)*Feature Fusion*: Aggregates cross-domain data. The structure of the UDL interface is shown in Fig. 4.

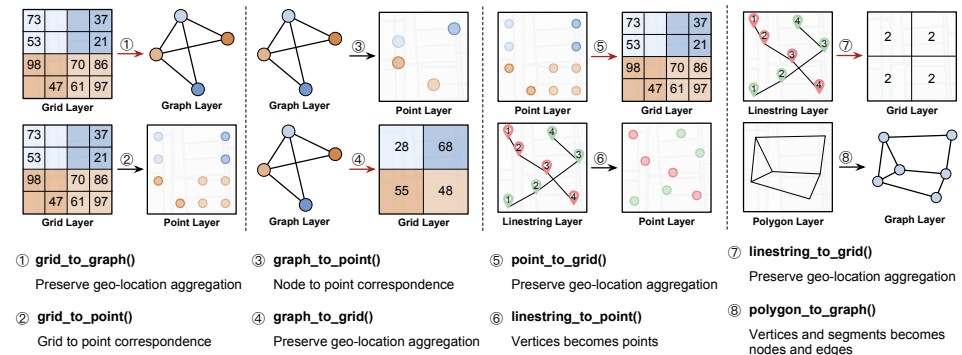

Figure 3: Transformation within layers. Red arrows indicate that there is intra-area aggregation during the transformation process, which may lose some precision.

## 3.3 Productivity

Data from diverse domains comprise numerous modalities, each recorded by distinct data types, distributions, scales, and granularities. For example, satellite images [13] are represented by pixel intensities, whereas POIs [39, 12] are usually represented by spatial points linked to a static category. Human mobility data [15] is embodied as trajectories, while road networks are represented as graph [18] and population data [21] is represented as grid-based data with real-value. The property of multiple data layers and friendly APIs of UDL well facilitates the combination of features, which

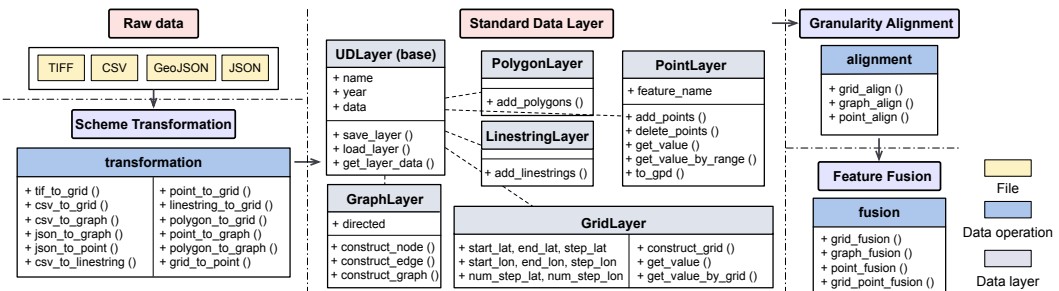

Figure 4: The design and structure of the UDL interface.

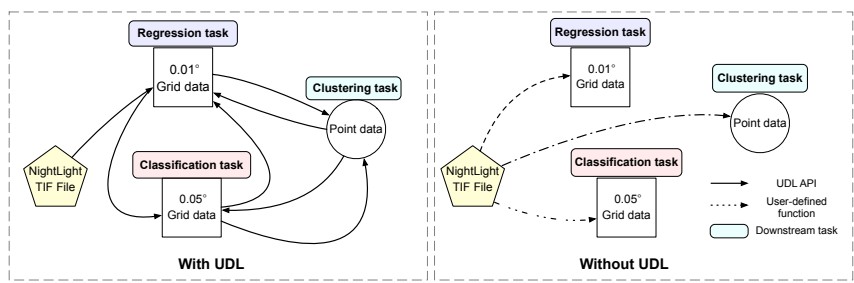

Figure 5: Implementing multiple downstream tasks with the same data through UDL data layers and unified APIs.

is evident in two aspects, as depicted in Fig. 5. First, the transition from diverse raw data sources to standard layer data of specified structures becomes routine and procedural, eliminating the need for repetitive processing (via user-defined functions) of similar data types. Second, various data layers can be quickly and easily aligned with or transformed to each other in UDL according to the geographic coordinate characteristics of urban data, rather than reprocessing from the original data every time. Both of their outputs can be directly used as inputs to the model or with minor adjustments.

Using the nightlight data from the three experimental cases described in later sections as an example, we include $0.02°$ and $0.01°$ grid data in Shanghai, $0.05°$ and $0.01°$ grid data in New York and point data. Without UDL, processing from the raw data needs to be conducted 6 times. However, with UDL, only 2 steps are required using the ready-to-use API. Subsequently, only 4 times exist between the UDL layer, where the users need to complete the conversion from 0 times.

Especially with the rise of large language models related to urban computing such as time-series foundation models, UDL facilitates easy data fusion for these models. To demonstrate this idea with an example of time-series foundation models UniTS [11] and PatchTST [31], various data types can be transformed into point data as inputs to both models. And this form is the main data provider for the current time-series models [50]. We anticipate that it will be a significant tool for urban-related large language models.

## 4 Empirical Cases

In this section, we use four typical downstream tasks to illustrate how UrbanDataLayer (UDL) can accelerate and enhance urban research. Four cases cover both supervised learning and unsupervised learning tasks, including $PM_{2.5}$ concentration prediction, built-up areas classification, identification of administrative boundaries, and El Nino anomaly detection. A more detailed description of data and implementation of cases are provided in `https://github.com/SJTU-CILAB/udl`.

Table 2: Effectiveness of combining different features in $PM_{2.5}$ prediction problems. The best performance of the combination for each compared method is underlined and the best performance of all is bolded. Overall, in $PM_{2.5}$ prediction, combining more features contributes to better performance.

| Region | Shanghai | | | | | | New York | | | | | |
|---|---|---|---|---|---|---|---|---|---|---|---|---|
| Method | XGBoost | | | MLP | | | XGBoost | | | MLP | | |
| Measurement | RMSE | MAE | $R^2$ | RMSE | MAE | $R^2$ | RMSE | MAE | $R^2$ | RMSE | MAE | $R^2$ |
| Roadnet Intersection Density | 3.953 | 4.861 | 0.181 | 3.710 | 4.779 | 0.199 | 0.608 | 0.778 | 0.368 | 0.720 | 0.898 | 0.040 |
| Nightlight | 4.327 | 5.127 | 0.089 | 4.821 | 5.859 | -0.204 | 0.743 | 0.936 | 0.084 | 0.721 | 0.881 | 0.074 |
| Population | 4.374 | 5.134 | 0.086 | 4.373 | 5.185 | 0.057 | 0.740 | 0.932 | 0.093 | 0.676 | 0.854 | 0.130 |
| Roadnet + Nightlight | 3.672 | 4.582 | 0.272 | 3.404 | 4.276 | 0.359 | 0.591 | 0.762 | 0.393 | 0.648 | 0.828 | 0.183 |
| Roadnet + Population | 3.669 | 4.535 | 0.287 | 3.464 | 4.365 | 0.332 | 0.591 | 0.764 | 0.390 | 0.677 | 0.864 | 0.111 |
| Nightlight + Population | 3.974 | 4.783 | 0.207 | 4.044 | 4.810 | 0.189 | 0.713 | 0.901 | 0.151 | 0.619 | 0.792 | 0.252 |
| Combining All | 3.355 | 4.235 | 0.378 | **3.103** | **4.075** | **0.417** | **0.578** | **0.753** | **0.408** | 0.644 | 0.817 | 0.204 |

- The major experiments are composed of results of combining various features using classic methods to demonstrate the benefit of unifying diverse data input via UDL. In this sense, innovating advanced methods for each task is not within our scope.

- Successfully conducting the experiments justifies the fact that: (1) UDL facilitates easy processing of data to build reproducible benchmarks; (2) UDL is applicable across different spatial regions, temporal periods, and feature dimensions, thereby enabling the scaling up of spatial-temporal data.

## 4.1 $PM_{2.5}$ concentration prediction

Accurate air quality prediction is of great importance to urban governance and human livelihood [12]. In this paper, we study the frequently-discussed $PM_{2.5}$ concentration prediction problem [20, 22, 26]. We use XGBoost and MLP models, combining night-time lights, population, and road intersection density as inputs, to conduct experiments in Shanghai, China ($120°E \sim 122°E,\ 30°N \sim 32.4°N$) and New York State, United States ($80°W \sim 70°W,\ 40°N \sim 45.5°N$). The predicted results are evaluated against the value obtained from the NASA Socioeconomic Data and Applications Center (recognized as ground truth on all grids). Three metrics are considered respectively as RMSE, MAE and $R^2$. The data is split into training data and test data at a ratio of 9:1. Table 2 shows the performance of different feature combinations on $PM_{2.5}$ concentration prediction in two regions. It is observed that combining more features performs better on XGBoost and MLP overall. The observed patterns can be attributed to the strong spatial correlation between intersection density, nightlight, population, and the $PM_{2.5}$ (as shown in Fig. 6 and Fig. 7). The figures depict grid aggregation, where each cell value represents the average of the original values within that cell. The granularity of the data is $0.02° \times 0.02°$ per grid in Shanghai, and $0.05° \times 0.05°$ per grid in New York State. An interesting observation is that areas with higher values for the three urban features—intersection density, nightlight, and population—tend to exhibit higher $PM_{2.5}$ concentrations, as seen in New York City and downtown Shanghai. These results indicate that incorporating knowledge from more domain-relevant data sources enhances the accuracy of environmental pollution predictions.

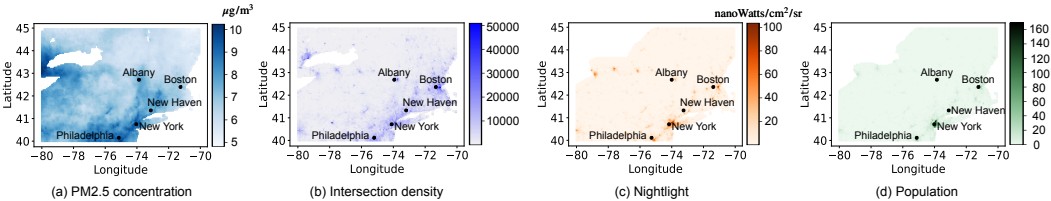

(a) PM2.5 concentration    (b) Intersection density    (c) Nightlight    (d) Population

Figure 6: $PM_{2.5}$, intersection density, night-time light intensity, and population density in areas near New York State. Big cities, e.g., New York and Boston, exhibit high values across all four dimensions. Among these urban features, intersection density emerges as the most significant factor in predicting $PM_{2.5}$ concentrations.

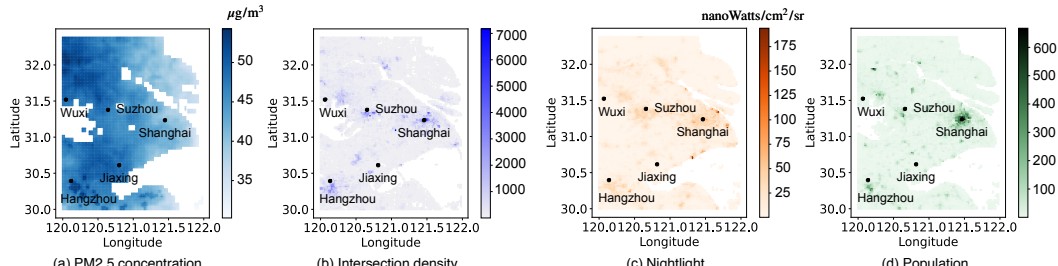

Figure 7: Urban data of $PM_{2.5}$ prediction task in Shanghai. Missing values are imputed by the mean along each column. Values have the same meaning as in Fig. 6.

Table 3: Effectiveness of combining different features in built-up surface classification problems. The best performance of the combination for each compared method is underlined and the best performance of all is bolded.

| Region | | Shanghai | | | | | New York | | | | |
|---|---|---|---|---|---|---|---|---|---|---|---|
| Method | | LR | DT | RF | GBDT | Adaboost | LR | DT | RF | GBDT | Adaboost |
| **Accuracy** | Nightlight | 0.736 | 0.653 | 0.653 | 0.747 | 0.744 | 0.783 | 0.733 | 0.734 | 0.808 | 0.808 |
| | SMOD | 0.764 | 0.767 | 0.768 | 0.767 | 0.760 | 0.729 | 0.729 | 0.729 | 0.729 | 0.729 |
| | Population | 0.761 | 0.677 | 0.677 | 0.767 | 0.766 | 0.861 | 0.818 | 0.818 | 0.869 | 0.868 |
| | Nightlight+SMOD | 0.781 | 0.706 | 0.715 | 0.786 | 0.782 | 0.862 | 0.820 | 0.821 | 0.871 | 0.869 |
| | Nightlight+Population | 0.781 | 0.708 | 0.710 | 0.786 | 0.782 | 0.862 | 0.820 | 0.821 | 0.871 | 0.869 |
| | SMOD+Population | 0.781 | 0.707 | 0.732 | 0.786 | 0.782 | 0.862 | 0.820 | 0.823 | 0.871 | 0.869 |
| | All | 0.782 | 0.714 | 0.772 | **0.794** | 0.790 | 0.863 | 0.819 | 0.857 | **0.873** | 0.871 |
| **F1** | Nightlight | 0.710 | 0.663 | 0.664 | 0.746 | 0.731 | 0.814 | 0.785 | 0.786 | 0.853 | 0.851 |
| | SMOD | 0.788 | 0.786 | 0.787 | 0.786 | 0.737 | 0.738 | 0.738 | 0.738 | 0.738 | 0.738 |
| | Population | 0.743 | 0.690 | 0.690 | 0.770 | 0.758 | 0.884 | 0.853 | 0.854 | 0.896 | 0.894 |
| | Nightlight + SMOD | 0.777 | 0.718 | 0.725 | 0.791 | 0.792 | 0.884 | 0.855 | 0.856 | 0.897 | 0.895 |
| | Nightlight + Population | 0.777 | 0.719 | 0.721 | 0.791 | 0.792 | 0.884 | 0.855 | 0.856 | 0.897 | 0.895 |
| | SMOD + Population | 0.777 | 0.719 | 0.739 | 0.791 | 0.792 | 0.884 | 0.855 | 0.858 | 0.897 | 0.895 |
| | All | 0.776 | 0.725 | 0.778 | **0.800** | 0.793 | 0.886 | 0.854 | 0.886 | **0.899** | 0.896 |
| **AUC-ROC** | Nightlight | 0.742 | 0.652 | 0.653 | 0.749 | 0.748 | 0.789 | 0.717 | 0.717 | 0.780 | 0.784 |
| | SMOD | 0.760 | 0.765 | 0.765 | 0.765 | 0.765 | 0.765 | 0.765 | 0.765 | 0.765 | 0.765 |
| | Population | 0.765 | 0.677 | 0.677 | 0.768 | 0.769 | 0.864 | 0.807 | 0.807 | 0.858 | 0.860 |
| | Nightlight + SMOD | 0.784 | 0.706 | 0.715 | 0.787 | 0.782 | 0.865 | 0.809 | 0.810 | 0.859 | 0.860 |
| | Nightlight + Population | 0.784 | 0.707 | 0.710 | 0.787 | 0.782 | 0.865 | 0.809 | 0.809 | 0.859 | 0.860 |
| | SMOD + Population | 0.784 | 0.707 | 0.733 | 0.787 | 0.782 | 0.865 | 0.809 | 0.811 | 0.859 | 0.860 |
| | All | 0.784 | 0.714 | 0.773 | **0.795** | 0.790 | **0.866** | 0.807 | 0.845 | 0.861 | 0.863 |

## 4.2 Built-up areas classification

Obtaining accurate information about urban built-up areas is crucial for urban planning and management [34]. In this paper, we investigate the problem of using population, nightlight, and urban index to classify the urban region functions in the level of $0.01° \times 0.01°$ in space. The experimental areas of interest are Shanghai and New York State, consistent with the previous section. Five classic classifiers are chosen for this task: Logistic Regression (LR), Decision Tree (DT), Random Forest (RF), Gradient Boosting Decision Tree (GBDT) [10] and AdaBoost [9]. To verify the feasibility of the combination, the accuracy, F1-score (the average harmonic mean of precision and recall), and the Area Under the Curve (AUC) of the Receiver Operating Characteristic (ROC) are used as the main metric for the classification tasks.

The results are shown in Table 3, from which we have the following observations. (1) Combining all features achieves the best performance in both regions, which means the nightlight, SMOD (an indicator showing the degree of urbanization), and population all contribute to the identification of built-up areas. (2) By further analyzing the SHAP value, we demonstrate the impact of each feature for individual samples. As observed in Fig. 8 (e), SMOD has more total impact than the other two features, while for some regions nightlight matters much more. Relation within the data also garners considerable attention. As depicted in Fig. 8 (a) - (c), SMOD values have a more positive impact on classification when both SMOD and population values are high in the region. SMOD tends to be higher when the population is higher, which collectively causes a positive influence. When nightlight values are the same, the lower the SMOD, the more positive the effect they have on classification.

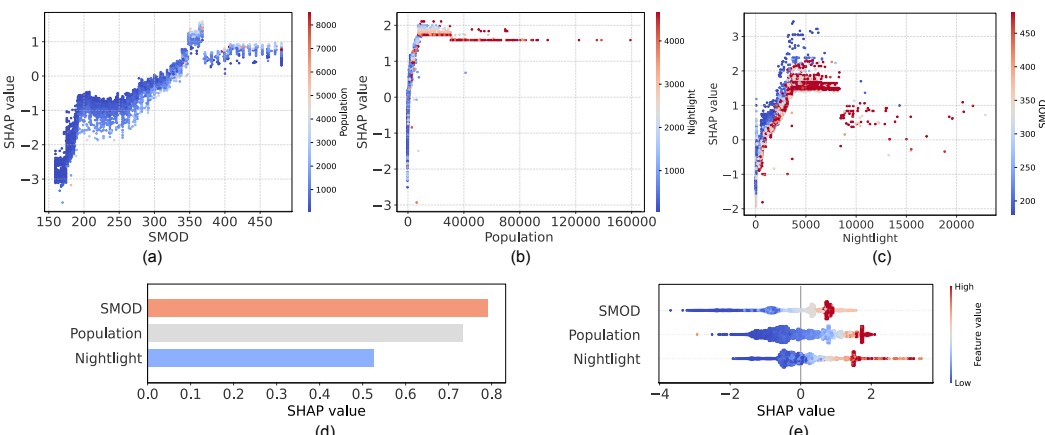

Figure 8: SHAP value analysis of three features for built-up areas classification in Shanghai. (a) - (c) illustrate the interactions between each feature and other features, where each data point represents a sample. In (d), SMOD has the highest mean SHAP value across all given samples, indicating it has the most influence on the results. (e) presents the SHAP values under each feature value, with color representing the level of the feature value.

## 4.3 Identification of administrative boundaries

Identifying the boundaries of cities is crucial for urban planning (e.g., infrastructure building) and urban service arrangement (e.g., delivery). It is believed that using human activity data, e.g., POI, population, road network data, and nightlight data, can help to identify the city boundary. By utilizing UDL to unify the aforementioned data to point-wise data, this task can be further formulated as a clustering problem. Two commonly used clustering methods, K-Nearest Neighbor (KNN) and Gaussian Mixture Model (GMM) are used, and the clustered boundaries are compared with public administrative district boundaries. Here, we consider two metrics of this specific task. (1) F1-score: The F1-score is the harmonic mean of precision and recall. Precision focuses on the number of points assigned to a district that actually belong to that distinct while recall is more concerned with how many points belonging to a district are successfully clustered. (2) IOU: We calculate the Intersection over Union (IOU) between the obtained clustering boundaries and corresponding administrative districts.

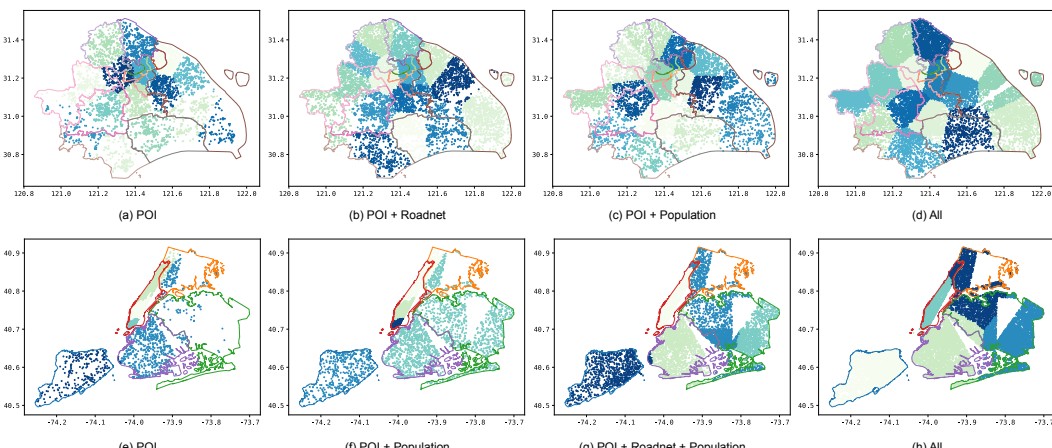

Figure 9: Clustering results in Shanghai using K-means Model ((a) - (d)) and in New York City using Gaussian Mixture Model ((e) - (h)). The x and y coordinates represent latitude and longitude respectively. The points of different colors indicate different clusters predicted, and the polygons of different colors are the ground truth of the administrative divisions.

Table 4: Effectiveness of combining different features in boundary identification problems. The best performance of the combination for each compared method is underlined and the best performance of all is bolded.

| City | Shanghai | | | | New York City | | | |
|---|---|---|---|---|---|---|---|---|
| Method | KNN | | GMM | | KNN | | GMM | |
| Measurement | F1 | IOU | F1 | IOU | F1 | IOU | F1 | IOU |
| POI | **0.608** | 0.338 | 0.571 | 0.300 | 0.557 | 0.207 | 0.625 | 0.303 |
| POI + Roadnet | 0.542 | **0.349** | 0.497 | 0.332 | 0.867 | 0.330 | 0.874 | 0.511 |
| POI + Nightlight | 0.499 | 0.329 | 0.479 | 0.294 | 0.864 | 0.341 | 0.771 | 0.411 |
| POI + Population | 0.455 | 0.231 | 0.463 | 0.228 | 0.467 | 0.281 | 0.542 | 0.306 |
| POI + Roadnet + Nightlight | 0.489 | 0.319 | 0.473 | 0.315 | 0.891 | 0.370 | 0.706 | 0.434 |
| POI + Nightlight + Population | 0.509 | 0.309 | 0.435 | 0.279 | 0.874 | 0.350 | 0.702 | 0.398 |
| POI + Roadnet + Population | 0.471 | 0.286 | 0.463 | 0.309 | **0.914** | 0.367 | 0.913 | 0.577 |
| Combining All | 0.475 | 0.327 | 0.399 | 0.280 | 0.899 | 0.376 | 0.909 | **0.613** |

Table 5: Effectiveness of combining different features in anomaly detection problems. The best performance of each compared method is bolded and the second best performance is underlined.

| | Method | EI Nino Dataset | | | | | |
|---|---|---|---|---|---|---|---|
| | | LOF | CoLA | ANOMALOUS | GAE | OCGNN | ONE |
| AUC-ROC | SP[1]+ ZW[2]+ MW[3] | 0.525 | 0.540 | **0.469** | 0.489 | **0.498** | 0.469 |
| | SP + Humidity + AT[4] | 0.522 | 0.450 | 0.463 | 0.482 | 0.496 | 0.464 |
| | SP + ST[5]+ AT | 0.525 | **0.542** | 0.466 | 0.488 | 0.493 | **0.476** |
| | SP + ZW + MW + Humidity + AT | **0.540** | 0.440 | 0.456 | **0.499** | 0.495 | 0.457 |
| | All | 0.538 | 0.425 | 0.449 | 0.478 | 0.507 | 0.463 |

[1] SP: Spatial information contains longitude and latitude.
[2] ZW: Zonal winds (west $<$ 0, east $>$ 0).
[3] MW: Meridional winds (south $<$ 0, north $>$ 0).
[4] AT: Air temperature.
[5] ST: Sea surface temperature and subsurface temperatures down to a depth of 500 meters.

From Table 4, we observe that using POI information alone achieves the best performance in Shanghai while adding auxiliary data yields better results in New York City. Fig. 9 provides insight into this difference: in Shanghai, POI data effectively differentiates between urban and suburban areas, while the population and road network data distribute more evenly across various districts, which can compromise the distinguishing capability of POI data. Conversely, in New York City, POI data alone is insufficient, and the addition of auxiliary data complements the POI information, leading to improved performance.

## 4.4 El Nino anomaly detection

Detecting urban anomalies (e.g., traffic anomaly, unexpected crowds, environment anomaly, and individual anomaly) holds significant importance in the endeavor to enhance the urban life quality and arrange emergency actions [44]. Here, we use El Nino dataset as an example to demonstrate how UDL assists in outlier detection tasks. The original dataset is assumed to be without anomalies. Following the approach in [8], we introduce anomalies constituting 2% of the dataset. We then compare the performance of different combinations of node features using various anomaly detection methods [24], including LOF [4], CoLA [25], ANOMALOUS [33], GAE [16], OCGNN [38] and ONE [3]. The evaluation metric utilized is the Area Under the Curve (AUC) of the Receiver Operating Characteristic (ROC).

We show AUC values for all methods on all feature combinations in Table. 5. It is observed that the combination of spatial information, zonal winds, and meridional winds achieves relatively better results overall. The best combination of results is sea surface temperature and air temperature using CoLA. Moreover, we shed light on some interesting observations regarding the results to explain why it is more likely to be an outlier. In Fig. 10 (a), considering the properties of air temperature and sea surface temperature, the outlier is similar to its neighbors in one of the attributes while another is much higher or lower. We can observe that the detected anomaly's (upper left) air temperature is around $26°$. But its sea surface temperature is higher than $28.5°$ where its "neighboring" samples with the same air temperature are below $28°$. Similar observations in structural aspects can be made

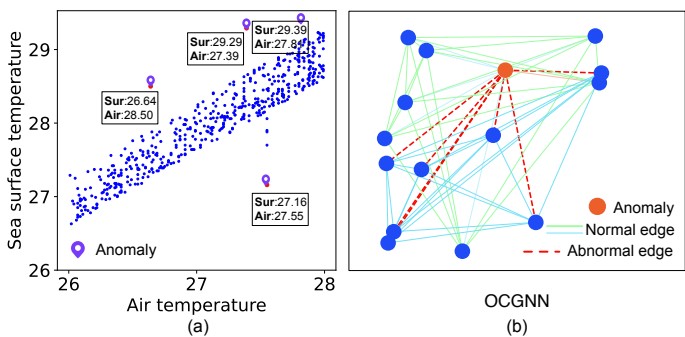

Figure 10: The detected anomaly and surrounding points of El Nino region. (a) Detected anomaly points by CoLA. (b) Detected structural anomaly nodes by OCGNN.

in Fig. 10 (b), where an anomaly may be a node whose edges are inaccurately linked. As the edges are established based on spatio-temporal information, edge relationships exist between nodes that have the same temporal or spatial information. The node in the graph is recognized as an anomaly because the spatio-temporal feature is replaced, making the edges in the dotted line unusually present. Since the features of the anomalies are replaced randomly, the best combination of features may be stochastic.

# 5   Conclusion and Outlook

This paper introduces a unified data pipeline including standard data structures and easy-to-use processing interfaces on urban research. We define the standard data layers from five common data organizations used in urban science and provide three components in the pipeline. UDL mitigates the gap between various urban data and urban computing research by addressing the challenges: (1) handling dirty and repetitive data processing, (2) establishing a unified standardized format, (3) integrating alignment and fusion for urban data. This will enable reproducible benchmark construction and foster the development of the multi-modal databases. The effectiveness and productivity of UDL have been demonstrated in four instances. We believe it will become a promising data tool to inspire more researchers to tackle the urban problems our cities face.

When data layers are constructed globally, the availability of sufficient data facilitates large-scale urban research and the development of large models [5, 14]. Despite the high productivity of UDL, the alignment of urban data is currently limited to geospatial information and future research could explore more aspects. In the future, we will incorporate tasks across regions and explore solutions to urban issues on a global scale.

## Acknowledgments and Disclosure of Funding

This work was sponsored by National Natural Science Foundation of China under Grant No. 62102246, 62272301, and Provincial Key Research and Development Program of Zhejiang under Grant No. 2021C01034. Part of the work was done when the students were doing internships at Yunqi Academy of Engineering.

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
