# UrbanDataLayer: A Unified Data Pipeline for Urban Science

**Yiheng Wang** [1]**, Tianyu Wang** [1]**, Yuying Zhang** [1]
**Hongji Zhang** [1]**, Haoyu Zheng** [1]**, Guanjie Zheng**[*,1]**, Linghe Kong** [*,1]
[1] Shanghai Jiao Tong University, Shanghai, China
{yhwang0828, wty500, shjtzyy01, zhanghongji
langanzheng, gjzheng, linghe.kong}@sjtu.edu.cn

## 1 Detailed Introduction of Urban Data Layer Components

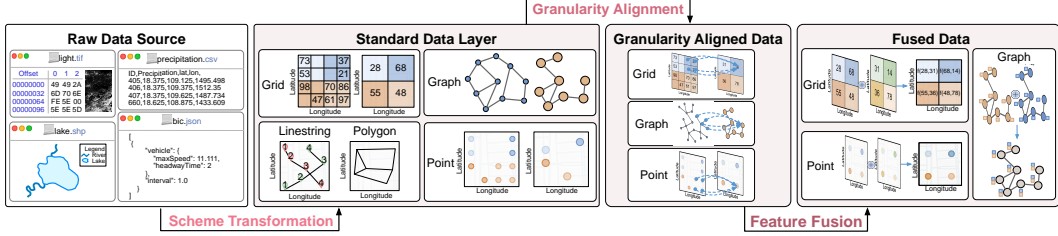

Figure 1: The components of UrbanDataLayer. The words in red are the data processing steps.

The overall procedure of UDL data processing is shown in Fig. **1**. It can be viewed as four stages of data wrappers (in black bold characters) divided by three data processing steps (in red characters).

### 1.1 Four Data Wrappers

The four data wrappers contain a series of data intermediate states from raw data to fused data which can be directly used by the models.

(1) **Raw data source.** The raw data refers to unprocessed source data obtained directly from sensors or geographically relevant data obtained from the network such as raster data. These data is usually stored in various forms of structured files such as JSON, CSV and several image files. Raw data presents different structures and must undergo data pre-processing operations before being fed into the model.

(2) **Standardized data layer.** The standardized data layer is a uniformly defined data structure and consists of different types of urban data. It contains both data spatio-temporal information and the ready-to-use data itself. We define five types of data layer, including grid data, graph data, point data, linestring data and polygon data.

- Grid data: Grid data represents the city data at a specific range of latitude and longitude and at a certain granularity. The concerned area is partitioned into $I \times J$ grids according to the geographical coordinates. Given urban data $D = \{d_{i,j}, 0 \le i \le I-1, 0 \le j \le J-1\}$, $d_{i,j}$ indicates the feature value within the latitude and longitude range to which the grid $(i, j)$ belongs.

---

[*]Corresponding Author.

Submitted to the 38th Conference on Neural Information Processing Systems (NeurIPS 2024) Track on Datasets and Benchmarks. Do not distribute.

Table 1: Summary of data formats. The common methods of saving, loading and getting data are omitted.

| Type | Properties | Methods |
|------|-----------|---------|
| Graph layer[1] | $\underline{name}$,[2] lon, lat, directed, year, data | construct_node, construct_edge, construct_graph |
| Grid layer | name, start_lat, end_lat, step_lat, start_lon, end_lon, step_lon, year, data | construct_grid, get_value, get_region, get_value_by_grid, print_info, save_grid |
| Point layer | name, feature_name,[3] year, data | add_points, delete_points, get_value, get_value_by_range, to_gpd |
| Linestring layer | name, year, data | add_linestrings, delete_linestrings |
| Polygon layer | name, year, data | add_polygons, delete_polygons |

[1] The methods of NetworkX [1] are also available for the GraphLayer object.
[2] The underline is the key of the node features.
[3] The feature_name is the list of point features' types besides the main feature.

i$(i = 0, 1, ..., I - 1)$ and j$(j = 0, 1, ..., J - 1)$ denote the indice of latitude (row) and longitude (column), respectively. At this stage, each grid data layer stores a single feature.

- Graph data: Graph data constructs the data as a graph $G = (V, E, X)$. V is the set of nodes, E is the set of edges where each edge can be represented as $e_{ij} = (v_i, v_j)$ and X is the set of node features. The properties of the edges are optional. At this stage, each graph data layer stores individual node characteristics along with the latitude and longitude of node.

- Point data: Point data is defined as $P = \{p_i\}$, where each point $p_i = (x_i, y_i, \mathbf{v}_i)$ where $(x_i, y_i)$ stands for its geographical coordinate location and $\mathbf{v}_i$ stands for its features' value. It can also be denoted as $\mathbf{X}_i \in \mathbb{R}^C$ where $C$ is the number of features. Each point layer may contain multiple features at this stage.

- Linestring data: Linestring data is usually applied to the representation of trajectories in city. Data in linestring layer is $L = \{l_i\}$, where $l_i = [(x_0, y_0), (x_1, y_1), ..., (x_n, y_n)]$ describes the lines between the points. The elements in the list can also be the type of point data. Each layer can be consist of multiple linestrings.

- Polygon data: Polygon data $D_{plg} = d_i$ uses the points of a boundary to denote a linearly enclosed area and multiple areas cannot cross each other where $d_i = [(x_0, y_0), (x_1, y_1), ..., (x_m, y_m)]$. The holes bounded by linear rings in the polygon region can also be stated. A polygon data layer can contain multiple polygons.

The data formats of UDL vary between different types of data layers and also have some common properties. Each UDL class basically comprises the layer name (the meaning of the urban data, e.g., nightlight, population, etc.), the year (year of the recorded data) and the data itself. Due to the urban nature of UDL, all data layers have geographic information identified by latitude and longitude (can also be replaced with other unique geographic identifiers).

For the defined five types of UDL, data operations like constructing data, modifying data and querying data by coordinates are provided. Subtle differences exist between various classes of UDL and a uniform interfaces to save and load data is accessible. The exclusive properties and methods of each class are described briefly in Table 1. More details can be found in the document of UDL[1].

(3) **Granularity aligned data.** Granularity aligned data is processed from UDL layer data with different granularity through offered granularity transformation methods, specifying required spatial granularity. Data after granularity transformation differs from the original standardized layer data only in spatial granularity by some aggregation method, such as averaging, summing, and so on. Multi-source data of the same layer type can be consistent in both spatial coverage and granularity after this process.

(4) **Fused data.** Fused data is several aligned layer data fused from multiple urban data sources. Urban data is heterogeneous and the relationships between different domains also cannot be ignored [2]. Like in forecasting traffic flow, external factors including both weather conditions, temperature and wind speed were considered in [3]. Fused data is processed from granularity aligned data with the same granularity and then multi-source data can be integrated. The greater the variety of data types involved in a given task, the higher the probability of encountering a

[1] https://urbandatalayer-doc.readthedocs.io/en/latest/

data scarcity issue [4]. The obtained fused data is also in UDL data format, where values are concatenated or aggregated in a specified way.

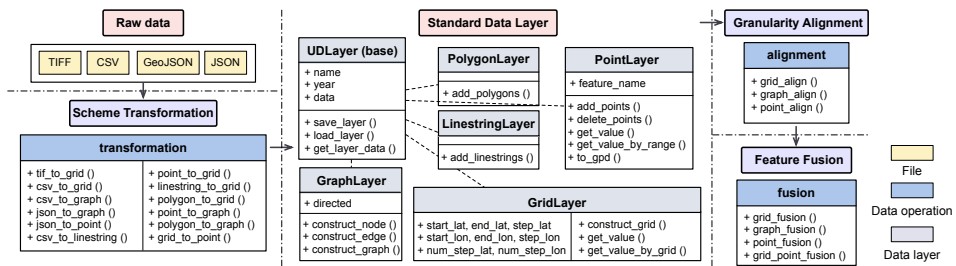

Figure 2: The design and structure of UDL interface. The raw data file format can be directly transformed into UDL through *Scheme Transformation* as some layers receiving files to initialize.

## 1.2 Three Data Processing Steps

The UrbanDataLayer builds the data layers and user-friendly APIs that make it easier to process and reuse city data in urban research, provided with both codes and some city data. The overall structure and processing flow are shown in Fig. 2. And the processing operations of UrbanDataLayer between four data wrappers are as follows.

(1) **Scheme Transformation.** Scheme transformation consists of two types: from raw data source to standard data layer and between data layers. For the type from raw data, methods automatically transforming raw data into UDL layer data are provided. Structured data handling is considered, such as CSV files, TIFF files and JSON files, etc. For instance, `tif_to_grid()` interface transforms a TIFF file to a grid layer data with customized scope and granularity. We design this pipeline to address the challenge for researchers to go through the complex data processing process repeatedly when using similar urban datasets and we attempt to define it as a unified data transformation paradigm. All raw files need to contain geographical location information. After this, all data is converted to harmonized standard layer data. The mutual conversion within data layers are also supplied. Here we provide eight transformations between UDL data layers as illustrated in Fig. 3. For instance, when transforming polygons to graphs such as `polygon_to_graph()`, vertices and line segments become nodes and edges, respectively. It should be noted that such a basic transformation may be lossy (marked in red arrows).

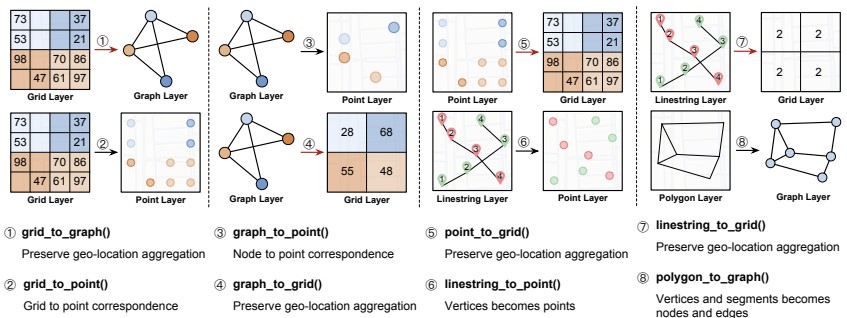

Figure 3: Transformation within layers. Red arrows indicate that there is intra-area aggregation during the transformation process, which may lose some precision.

(2) **Granularity Alignment.** Granularity alignment is a transformation among different granularities of data in the same data layer according to a specified aggregation, mainly considering the transformation from fine-grained data to coarse-grained data. Identical data sources may be processed at multiple granularities in different scenarios. Instantiated by a specific task, when

Table 2: Data used in cases and corresponding UrbanDataLayer types.

| Dataset | Region(s) | Type | Year[2] |
|---|---|---|---|
| Population | Shanghai, New York State | Grid | 2020 |
| Nightlight | Shanghai, New York State | Grid | 2016 |
| Built-up surface | Shanghai, New York State | Grid | 2020 |
| Roadnet intersection | Shanghai, New York State | Grid | 2022, 2023 |
| $PM_{2.5}$ | Shanghai, New York State | Grid | 2016, 2019 |
| SMOD | Shanghai, New York State | Grid | 2015 |
| POI | Shanghai, New York City | Point | 2022 |
| Boundary | Shanghai, New York City | Polygon | 2021, 2023 |
| El Nino | Equatorial Pacific | Graph[1] | 1989 - 1998 |

[1] Its original data type is Point.
[2] The two years indicate the year of data for two different regions, respectively.

predicting $PM_{2.5}$ concentrations for cities, it may be necessary to try different divisions of the city in order to find the most suitable grid units for prediction by `grid_align()`. This type of interface is designed mainly for grid, graph and point layers. We provide several optional aggregation methods and users can also define the aggregation function by themselves. Also, granularity alignment for multiple data source is a imperative operation before data fusion when faced with multi-sources data of varying granularities.

(3) **Feature Fusion.** Feature fusion mainly occurs mainly among identical UDL layer type with the same granularity. This interface can be accessed after the granularity alignment. Considering the solutions of urban tasks usually combine data fusion into the model and training process, especially learning a representation of the original features from diverse datasets through the utilization of deep neural networks (DNN) [5, 2]. We provide aggregation and concatenation of cross-domain data which are most commonly used and easy to handle in the subsequent operations. We also provide a hybrid data fusion between point and grid `grid_point_fusion()` which is used in the Identification of administrative boundaries task.

In summary, the processing flow for urban data using UDL concludes as follows: (1) Convert to unified and standardized layer data with *scheme transformation*. (2) Transform to specified and aligned layer data by *granularity alignment*. (3) Complete necessary data fusion prior to the subsequent tasks through *feature fusion*.

## 1.3 Extensibility

We adopt a layered structure that is convenient for storing various types of city data. Data of different types, domains and times can be quickly inserted into the UDL, making it easy for all researchers to extend global urban data layer. Moreover, the unified layer structure allows the expeditious expansion of user-defined functions and the data output can be seamlessly accessed to other algorithms with high performance, helping subsequent researchers to reproduce it. The extensibility of UDL is promising to facilitate research on big data urban computing.

## 2 Data Access

Our UDL codebase is distributed under the MIT license. Users can easily access the tools through interfaces provided in `https://github.com/SJTU-CILAB/udl`.

# 3 Detailed Experiment Settings

## 3.1 Dataset Description

All datasets used in this paper are described here. All these datasets contain information on urban space. Data in each task have been aligned. The data used in this paper is also provided as ready-to-use layers as listed in Table 2.

- *Population*: This dataset from WorldPop records the population counts [6] of each unit. The original data granularity is $100\text{m} \times 100\text{m}$.

- *Night-time light*: This VIIRS night-time lights data is radiation value measured in $\text{nanoWatts}/\text{cm}^2/\text{sr}$ from WorldPop [7]. The original data source comes from NOAAs National Centers for Environmental Information, Visible Infrared Imaging Radiometer Suite (VIIRS). The original data granularity is $100\text{m} \times 100\text{m}$.

- $PM_{2.5}$: This dataset [8] provides an annual global $0.01°$ surface of concentrations ($\mu\text{g}/\text{m}^3$) of all composition ground-level fine particulate matter of 2.5 micrometers or smaller ($\text{PM}_{2.5}$).

- *Built-up surface*: The built-up surface ($\text{m}^2$) is the gross surface (including the thickness of the walls) bounded by the building wall perimeter, which from Global Human Settlement Layer (GHSL).

- *SMOD*: This dataset from GHSL indicates the Degree of Urbanisation by delineating and classifying settlement typologies via a logic of cell clusters population size, population and built-up area densities.

- *POI*: The POIs (point of interest) of Shanghai and New York City are cleaned from Baidu Maps [9] and Safe Graph [10], respectively.

- *Roadnet intersection*: The road network intersection data are collected from OpenStreetMap [11], representing the number of intersections in each unit.

- *Districts boundary*: The Shanghai administrative districts boundary is exported from DataV [12] and New York City from NYC Open Data [13].

- *El nino*: The El Nino dataset is provided by UCI repository [14] and we use a subset of the data provided by [15] containing 93,935 samples, measuring oceanographic and surface meteorological variables.

Not all data were updated to the same latest year, and we choose updated latest real data over predicted data. Since the data used is considered to be relatively static in the city, it does not fluctuate significantly over a few years.

## 3.2 Four Empirical Cases Detailed Settings

### 3.2.1 $\text{PM}_{2.5}$ concentration prediction.

Grid layer type is used in this case. The granularity of the data is $0.02° \times 0.02°$ per grid in China, and $0.05° \times 0.05°$ per grid in New York State. The grid aggregation method is an average of the original values. Missing values in data are replaced by the mean along each column. We treat each grid as a individual sample in this case. The data is split into training data and test data at a ratio of 9:1.

### 3.2.2 Built-up areas classification.

The data used in classification are all organized in grid layer format and well-aligned in granularity $0.01° \times 0.01°$ in space. As a classification task, we convert the built-up surface which is continuous data into boolean values by setting a threshold (in Shanghai the threshold is set as 90000, while in New York State it is 1500) to decide which grid is an urban area so that the proportion of positive and negative samples is close. The data is split into training data and test data at a ratio of 8:2.

### 3.2.3 Identification of administrative boundaries.

The given multi-modal data including POIs (point of interest), road intersection density, population and night-time light consists of both point data and grid data. The feature value of point type data POI is its site category here, and the other three are grid data of $0.05° \times 0.05°$ granularity for both city. Note that, we only consider the coordinates of POIs and sample grid feature data as coordinate points $(x, y)$ according to the grid value to perform data fusion. For POI data, we randomly sample $\frac{1}{100}$ coordinate points from Shanghai and $\frac{1}{10}$ from New York City due to different POI density. Here, 5749 points remain in Shanghai and 11311 points remain in New York City. For other three grid data, we randomly generate coordinate points in each grid from their specific distribution determined by grid value. In each grid $(i, j)$, the point number

$$|P(i,j)| = \frac{d_{i,j}}{\max\limits_{\substack{0 \le s \le I-1 \\ 0 \le t \le J-1}} d_{s,t}} \cdot k \tag{1}$$

$d_{i,j}$ indicates the feature value within the latitude and longitude range to which the grid belongs. To be noted, the highest grid value is selected among New York State. $k$ is 1000 for Shanghai and 10000 for New York City. After that, we mix all the generated coordinate points as the fused data. The number of clusters is set according to the administrative districts, where Shanghai is set to 15 and New York City is set to 5. Chongming district of Shanghai is excluded due to the lack of POI data.

### 3.2.4 El Nino anomaly detection.

The data before processing is tabular data and consists of the following variables: date, latitude, longitude and zonal winds etc. twelve attributes. First we convert the date to a standard timestamp and then build the graph $\mathcal{G}$ based on the 93,935 sample. Each sample in the data are constructed as a node instantiated by its own node attributes. The edge sets of $\mathcal{G}$ contains three types: temporal (on the same day), spatial (adjacent within the grid distribution) and spatio-temporal (both of the former two). The constructed graph $\mathcal{G}$ comprises 93,935 nodes and 56,687,915 edges.

Various anomaly detection methods [16] are used to compare the performance of different combinations of node features:

- LOF [17]: Local outlier factor (LOF) is a method for finding outliers in a multidimensional dataset. It takes a local instead of a global view on outliers and the degree of isolation depends on surrounding neighborhood.

- IF [18]: Isolation Forest (iForest) is a model-based method which builds an ensemble of iTrees for a given data set and finds anomalies as those instances having short average path lengths on the iTrees.

- CoLA [19]: Contrastive self-supervised Learning framework for Anomaly detection on attributed networks (CoLA) is the first contrastive self-supervised learning-based method for graph anomaly detection and can efficiently captures the local information of a node and its neighboring substructure by a novel type of contrastive instance pair. Especially it is friendly to large-scale networked data.

- ANOMALOUS [20]: ANOMALOUS innovatively optimize attribute selection and anomaly detection as a whole. It selects representative instances based on CUR decomposition and uses residual analysis to measure the normality.

- GAE [21]: Variational Graph Auto-Encoders (GAE) is a framework for unsupervised learning on graph-structured data which makes use of latent variables and is capable of learning interpretable latent representations for undirected graphs.

- OCGNN [22]: One Class Graph Neural Network (OCGNN) is a novel framework, which combines the expressive power of Graph Neural Networks (GNNs) with the classical one-class objective. OCGNN outperforms other baseline methods in terms of anomaly detection accuracy and robustness. By leveraging the inherent structure and features of the graph, OCGNN effectively captures and represents the complex relationships and patterns within the data.

- ONE [23]: A novel unsupervised network embedding algorithm, which is designed to handle attributed networks with outliers. ONE generates robust network embeddings to minimize the influence of outlier nodes.

## 4  Datasheet

This appendix provides a datasheet for the used dataset. The format of this datasheet was introduced in [24] and consolidates the motivation, creation process, composition, and intended uses of our dataset as a series of questions and answers.

### 4.1  Motivation

Q1 **For what purpose was the dataset created?** *Was there a specific task in mind? Was there a specific gap that needed to be filled? Please provide a description.*

Urban-related research lacks standard benchmarks with unified data format. The UDL data aims to fill the gap of absent universal urban data.

Q2 **Who created the dataset(e.g.,whichteam,researchgroup) and on behalf of which entity (e.g., company, institution, organization)?**

The authors of this work (from Shanghai Jiao Tong University) created the UDL data.

Q3 **Who funded the creation of the dataset?** *If there is an associated grant, please provide the name of the grantor and the grant name and number.*

This work was sponsored by National Natural Science Foundation of China under Grant No. 62102246, 62272301, and Provincial Key Research and Development Program of Zhejiang under Grant No. 2021C01034. Part of the work was done when the students were doing internships at Yunqi Academy of Engineering.

Q4 **Any other comments?**

No.

### 4.2  Composition

Q5 **What do the instances that comprise the dataset represent (e.g., documents, photos, people, countries)?** *Are there multiple types of instances (e.g., movies, users, and ratings; people and interactions between them; nodes and edges)? Please provide a description.*

UDL data is composed of three components: (1) five base classes of UDL data; (2) a series of data process APIs to allow data transformation, alignment and fusion; (3) available UDL data for empirical urban cases.

Q6 **How many instances are there in total (of each type, if appropriate)?**

As for available UDL data for empirical cases in paper, there are six grid, one point, one polygon and one graph type UDL data regardless of granularity and region. However, UDL is a universal data conversion interface. Therefore, it will enable people to create more data instances using their own data.

Q7 **Does the dataset contain all possible instances or is it a sample (not necessarily random) of instances from a larger set?** *If the dataset is a sample, then what is the larger set? Is the sample representative of the larger set (e.g., geographic coverage)? If so, please describe how this representativeness was validated/verified. If it is not representative of the larger set, please describe why not (e.g., to cover a more diverse range of instances, because instances were withheld or unavailable).*

UDL sample dataset contain all possible type instance except linestring. Its large set lies in the expansion of spatial and temporal dimensions. The sample data can be any area and any time of available raw data.

Q8 **What data does each instance consist of? "Raw" data (e.g., unpro cessed text or images) or features?** *In either case, please provide a description.*

In the each type of UDL data, each value (per grid, node, point, etc.) represents a numerical value of urban characteristics at geographic location.

**Q9** **Is there a label or target associated with each instance?** *If so, please provide a description.*

No, we do not explicitly define a label or target for the instances.

**Q10** **Is any information missing from individual instances?** *If so, please provide a description, explaining why this information is missing (e.g., because it was unavailable). This does not include intentionally removed information, but might include, e.g., redacted text.*

Some value may be missing if the raw data is missing before transforming to the UDL data.

**Q11** **Are relationships between individual instances made explicit (e.g., users' movie ratings, social network links)?** *If so, please describe how these relationships are made explicit.*

The relationships between individual instances depend on raw data, which can be geographical adjacency, trajectory sequence, etc.

**Q12** **Are there recommended data splits (e.g., training, development/ validation, testing)?** *If so, please provide a description of these splits, explaining the rationale behind them.*

Not applicable.

**Q13** **Are there any errors, sources of noise, or redundancies in the dataset?** *If so, please provide a description.*

Since the data is transformed from various raw data in different formats and granularities, the data in UDL may have precision loss due to the geographic alignment. However, this totally depends on how the users of UDL put the settings.

**Q14** **Is the dataset self-contained,or does it link to or otherwise rely on external resources (e.g., websites, tweets, other datasets)?** *If it links to or relies on external resources, a) are there guarantees that they will exist, and remain constant, over time; b) are there official archival versions of the complete dataset (i.e., including the external resources as they existed at the time the dataset was created); c) are there any restrictions (e.g., licenses, fees) associated with any of the external resources that might apply to a dataset consumer? Please provide descriptions of all external resources and any restrictions associated with them, as well as links or other access points, as appropriate.*

UDL data relies on available open data source. a) It depends on the source data. However, since we aim to build urban research benchmark, it will remain constant when it comes to UDL format and can be reused. b) Yes. c) No.

**Q15** **Does the dataset contain data that might be considered confidential (e.g., data that is protected by legal privilege or by doctor patient confidentiality, data that includes the content of individuals' non-public communications)?** *If so, please provide a description.*

No.

**Q16** **Does the dataset contain data that, if viewed directly, might be offensive, insulting, threatening, or might otherwise cause anxiety?** *If so, please describe why.*

No.

**Q17** **Does the dataset relate to people?** *If not, you may skip remaining questions in this section.*

No.

**Q18** **Does the dataset identify any subpopulations (e.g., by age, gender)?** *If so, please describe how these subpopulations are identified and provide a description of their respective distributions within the dataset.*

No.

**Q19** **Is it possible to identify individuals (i.e., one or more natural persons), either directly or indirectly (i.e., in combination with other data) from the dataset?** *If so, please describe how.*

No.

Q20 **Does the dataset contain data that might be considered sensitive in any way (e.g., data that reveals race or ethnic origins, sexual orientations, religious beliefs, political opinions or union member ships, or locations; financial or health data; biometric or genetic data; forms of government identification, such as social security numbers; criminal history)?** *If so, please provide a description.*

No.

Q21 **Any other comments?**

No.

## 4.3 Collection Process

Q22 **How was the data associated with each instance acquired?** *Was the data directly observable (e.g., raw text, movie ratings), reported by subjects (e.g., survey responses), or indirectly inferred/derived from other data (e.g., part-of-speech tags, model-based guesses for age or language)? If the data was reported by subjects or indirectly inferred/derived from other data, was the data validated/verified? If so, please describe how.*

We gain the data from the open source mentioned in the supplementary, e.g, WorldPop and GHSL.

Q23 **What mechanisms or procedures were used to collect the data (e.g., hardware apparatuses or sensors, manual human curation, software programs, software APIs)?** *How were these mechanisms or procedures validated?*

We collect data mainly by two ways: (1) directly download the data file from the open website (e.g., tif and shape files); (2) use provided APIs to access raw data such as Baidu Map.

Q24 **If the dataset is a sample from a larger set, what was the sampling strategy (e.g., deterministic, probabilistic with specific sampling probabilities)?**

UDL data is not a sample from a larger set.

Q25 **Who was involved in the data collection process (e.g., students, crowdworkers, contractors) and how were they compensated (e.g., how much were crowdworkers paid)?**

Only the authors of this paper have been involved in the data collection process. No extra payment was made.

Q26 **Over what timeframe was the data collected? Does this timeframe match the creation timeframe of the data associated with the instances (e.g., recent crawl of old news articles)?** *If not, please describe the timeframe in which the data associated with the instances was created.*

The road network data was collected during December 2022. Other data was collected during 2023. They are not the latest data, but considering the inherent properties of urban they would not be changing a lot.

Q27 **Were any ethical review processes conducted (e.g., by an institutional review board)?** *If so, please provide a description of these review processes, including the outcomes, as well as a link or other access point to any supporting documentation.*

Not applicable.

## 4.4 Preprocessing/cleaning/labeling

Q28 **Was any preprocessing/cleaning/labeling of the data done (e.g., discretization or bucketing, tokenization, part-of-speech tagging, SIFT feature extraction, removal of instances, processing of missing values)?** *If so, please provide a description. If not, you may skip the remaining questions in this section.*

We aggregate the data at a specified geospatial granularity. The exact form and granularity of which are mentioned in the paper.

**Q29** **Was the "raw" data saved in addition to the preprocessed/cleaned/labeled data (e.g., to support unanticipated future uses)?** *If so, please provide a link or other access point to the "raw" data.*

Yes.

– https://www.worldpop.org/
– https://ghsl.jrc.ec.europa.eu/
– https://www.openstreetmap.org/
– https://data.cityofnewyork.us/City-Government/Borough-Boundaries/
  tqmj-j8zm
– https://www.safegraph.com/products/places
– http://datav.aliyun.com/portal/school/atlas/area_selector

**Q30** **Is the software that was used to preprocess/clean/label the data available?** *If so, please provide a link or other access point.*

Yes. The only software that was used to preprocess the data is Python, which is free to all users.

**Q31** **Any other comments?**

No.

### 4.5 Uses

**Q32** **Has the dataset been used for any tasks already?** *If so, please provide a description.*

Yes. We have presented four illustrative empirical cases, including $PM_{2.5}$ concentration prediction, built-up areas classification, identification of administrative boundaries and El Nino anomaly detection. See Section 4 in the main paper.

**Q33** **Is there a repository that links to any or all papers or systems that use the dataset?** *If so, please provide a link or other access point.*

N/A

**Q34** **What (other) tasks could the dataset be used for?**

Any urban-related tasks applying the standardized layer can use the data.

**Q35** **Is there anything about the composition of the dataset or the way it was collected and preprocessed/cleaned/labeled that might impact future uses? For example, is there anything that a dataset consumer might need to know to avoid uses that could result in unfair treatment of individuals or groups (e.g., stereotyping, quality of service issues) or other risks or harms (e.g., legal risks, financial harms)?** *If so, please provide a description. Is there anything a dataset consumer could do to mitigate these risks or harms?*

No.

**Q36** **Are there tasks for which the dataset should not be used?** *If so, please provide a description.*

No.

**Q37** **Any other comments?**

No.

### 4.6 Distribution

**Q38** **Will the dataset be distributed to third parties outside of the entity (e.g., company, institution, organization) on behalf of which the dataset was created?** *If so, please provide a description.*

No.

**Q39** **How will the dataset be distributed (e.g., tarball on websites, API, GitHub)?** *Does the dataset have a digital object identifier (DOI)?*

The data will be be found in GitHub.

**Q40** **When will the dataset be distributed?**

The sample data is already available.

**Q41** **Will the dataset be distributed under a copyright or other intel lectual property (IP) license, and/or under applicable terms of use (ToU)?** *If so, please describe this license and/or ToU, and provide a link or other access point to, or otherwise reproduce, any relevant licensing terms or ToU, as well as any fees associated with these restrictions.*

The UDL dataset will be distributed under the MIT license.

**Q42** **Have any third parties imposed IP-based or other restrictions on the data associated with the instances?** *If so, please describe these restrictions, and provide a link or other access point to, or otherwise reproduce, any relevant licensing terms, as well as any fees associated with these restrictions.*

No.

**Q43** **Do any export controls or other regulatory restrictions apply to the dataset or to individual instances?** *If so, please describe these restrictions, and provide a link or other access point to, or otherwise reproduce, any supporting documentation.*

No.

**Q44** **Any other comments?**

No.

## 4.7 Maintenance

**Q45** **Who will be supporting/hosting/maintaining the dataset?**

The authors.

**Q46** **How can the owner/curator/manager of the dataset be contacted (e.g., email address)?**

Contact the corresponding author or first author in the author list.

**Q47** **Is there an erratum?** *If so, please provide a link or other access point.*

N/A.

**Q48** **Will the dataset be updated (e.g., to correct labeling errors, add new instances, delete instances)?** *If so, please describe how often, by whom, and how updates will be communicated to dataset consumers (e.g., mailing list, GitHub)?*

Yes, we will keep polishing our processed data and everyone is encouraged to contribute new UDL data.

**Q49** **If the dataset relates to people, are there applicable limits on the retention of the data associated with the instances (e.g., were the individuals in question told that their data would be retained for a fixed period of time and then deleted)?** *If so, please describe these limits and explain how they will be enforced.*

Not applicable.

**Q50** **Will older versions of the dataset continue to be supported/hosted/maintained?** *If so, please describe how. If not, please describe how its obsolescence will be communicated to dataset consumers.*

Yes. Our UDL data has "year" attribute.

**Q51** **If others want to extend/augment/build on/contribute to the dataset, is there a mechanism for them to do so?** *If so, please provide a description. Will these contributions be validated/verified? If so, please describe how. If not, why not? Is there a process for communicating/distributing these contributions to dataset consumers? If so, please provide a description.*

Yes. everyone can release data in UDL format along with their research or contact us to add their UDL data in this work repository.

**Q52** **Any other comments?**

No.