# OpenReview forum: "UrbanDataLayer: A Unified Data Pipeline for Urban Science"
_NeurIPS.cc/2024/Datasets_and_Benchmarks_Track — NeurIPS 2024 Track Datasets and Benchmarks Poster_

### Official Review · Reviewer_phN3 · 2024-07-24
**Review for UrbanDataLayer**

**Rating:** 7
**Confidence:** 4
**Clarity:** Yes. The paper is well-written

**Review:**

Originality：This study identifies current problems in urban computing and urban science, and establishes a pipeline to transform raw data , granularity alignment, and feature fusion. UDL designed eight different types of conversions, which may have some loss of precision. Data passing through this pipeline is superior in the realization of downstream tasks. In four different typical case studies, UDL has successfully accomplished its task.

Significance： It is bentifical to build a large-scale benchmark and an effective multi-modal data fusion method . UDL is expected to be an effective tool to solve this problem, although the scope of investigation is limited to two metropolitan areas.

Clarity：The presentation of this work is largely clear, the reading basically fluent, and the logic largely reasonable.

Pros
Innovative Approach: The study introduces a novel pipeline for transforming raw data, granularity alignment, and feature fusion.

Practical Validation: The method's effectiveness is demonstrated through four typical case studies.

Relevance: Addresses pressing needs in urban computing and science as cities grow larger and generate more data.

Potential Impact: UDL could become a tool for large-scale benchmarks and multi-modal data fusion.


Cons

1) In the supplementary material, in part 1.2(2), the narrative on granularity alignment is too generalized, please provide details.

2) In the supplementary material, in part 1.2(3), The number of layers and other parameters of the DNN are not clearly stated.

3) Although UDL is essentially a multi-modal data fusion tool, the authors did not clearly explain its ability to fuse multi-modal data.

4) The four typical downstream tasks are based on cosmopolitan cities, and the authors need to clarify whether UDL works equally well for small areas with few data, and lots of dirty data and blank data.

**Strengths:**

1) Innovative Data Transformation Pipeline: The paper introduces a comprehensive and novel pipeline for transforming raw urban data through granularity alignment and feature fusion. T

2) Practical Case Studies: The method's effectiveness is validated through four typical case studies in different urban environments.

3) Addressing Critical Urban Challenges: The study identifies and tackles pressing issues in urban computing and urban science, particularly the need for a large-scale benchmark and effective multi-modal data fusion methods. This relevance to current urbanization trends underscores the paper's significance and potential impact on the field.

**Additional Feedback:**

Please refer to the  Opportunities For Improvement.

**Correctness:**

Yes, the evaluation methods and experiment design are appropriate and were performed correctly.

**Documentation:**

Yes. it is sufficient to support reproducibility

**Limitations:**

Yes, the limitation is discussed.

**Opportunities For Improvement:**

1) It is better to clearly explain the ability of the proposed tool to fuse multi-modal data.

2) In the current manuscript, the four typical downstream tasks are based on cosmopolitan cities. If possible, the authors should clarify whether UDL works equally well for small areas with sparse data, as well as areas with significant amounts of dirty data and blank data.

**Relation To Prior Work:**

Yes, the paper clearly distinguishes its contributions from prior work.

**Summary And Contributions:**

Currently, the development of urban science and urban computing faces two problems: the lack of standards due to the inconsistency of data processing steps, and the differences in multi-modal data formats that hinder data fusion. UrbanDataLayer (UDL) is proposed, which is a set of engineered models to help researchers to easily build up uniform standards and combine multi-modal data. To validate the effectiveness of UDL, the authors present four distinct urban problem tasks utilizing the proposed data layer, including PM2.5 concentration prediction, built-up areas classification, identification of administrative boundaries and El Nino anomaly detection. These examples prove that UDL is a perspective tool.

---

> ### Author Rebuttal · Authors · 2024-08-17
>
> Thanks for your insightful review. Here, we have carefully read the reviews and try to address the concerns as follows.
>
> > 1. In the supplementary material, in part 1.2(2), the narrative on granularity alignment is too generalized, please provide details.
>
> We will refine our explanation of this process in the camera-ready version. **For UDL grid layers, alignment occurs through these steps:**
>
> - Calculate the coordinate of new grids, according to the given step size.
>
> - For each new grid, find its nearest old grid.
>
> - Apply the old grid's value to the new grid.
>
> When the new grid's step is an integral multiple of the old grid's step, we can employ various fusion methods (e.g., "concat", "sum", "avg", "max", "min", or user-defined functions). For instance, when merging exactly four grids into one, averaging might yield better results than simply using the nearest old grid's value. When the new grid's step is not an integral multiple of the old grid's step, more sophisticated	interpolation tricks are involved. We will keep improving this.
>
> **Granularity alignment of UDL graph layers is conducted in the following steps:**
>
> - For each point in the graph, project it onto a grid with the designated step.
>
> - If two or more points fall into the same grid, fuse them. Otherwise, use the original point coordinate.
>
> - Reconnect the edges according to the original graph.
>
>
>
> ---
>
> > 2. In the supplementary material, in part 1.2(3), The number of layers and other parameters of the DNN are not clearly stated.
>
> For the MLP in PM2.5 prediction problem, we employ a straightforward structure:
>
> - Linear(num_features to 64),
> - ReLU,
> - Linear(64 to 512),
> - ReLU,
> - Linear(512 to output_num_features)
>
>  We used a random_state of 42, a learning rate of 1e-3, and trained for 50 epochs. Training Set : Test Set = 0.9 : 0.1
>
>
>
> ---
>
> > 3. Although UDL is essentially a multi-modal data fusion tool, the authors did not clearly explain its ability to fuse multi-modal data.
>
> UDL supports multi-modal inputs, including images, time series, and text, though some additional data preprocessing may be required.
>
> **In urban data science problems discussed in this paper, spatial and temporal information are usually the key. For those data that are completely unrelated with locations, they are out of scope for this paper.** However, for data that do relate to spatial or temporal aspects, we propose the following approaches:
>
> - **Image data:** each image is represented as a vector (containing its location and value), forming a grid where values are aggregated from all images in the grid. For time-varying images, this creates a multi-layer grid.
> - **Time series data:** processed as point data, with time points serving as attributes.
> - **Text data:** Text with hashtags is treated as a vector containing location and value, and processed in a similar way as image data. As for plain text content, a future direction is to employ Large Language Models (LLMs) to extract grid-wise information from the context.
>
> **It is important to note that associating data with specific locations inherently involves transforming that data into a UDL-compatible layer, such as a grid, graph, point, or polygon.** This is because spatial information must be representable on a map, and UDL encompasses these common types of spatial layers.  We can accommodate user-defined intermediate processing steps to be passed as function pointers, but the final output must still conform to one of the defined UDL layers to ensure consistency. Furthermore, should new multi-modal spatial representations emerge, we are open to expanding the UDL to include additional layer types, thereby enhancing its applicability to a wider range of urban data scenarios.
>
>
>
> ---
>
> > 4.  The four typical downstream tasks are based on cosmopolitan cities, and the authors need to clarify whether UDL works equally well for small areas with few data, as well as areas with significant amounts of dirty data and blank data.
>
> Yes. Many data we use are global, encompassing areas with sparse, dirty, or missing data. For example, in the boundary identification problem for New York City, POI data in Queens is largely missing. However, complementary data from roadnets, population and nightlight can compensate for this and yield a better classification result.
>
> Dealing with these data is an important research direction in urban science [1] [2] [3]. UDL can provide benefits and help various approaches to tackle these issues, including the use of complementary data layers, data imputation techniques, transfer learning, and other novel methods.
>
>
>
> References:
>
> [1] Ayush K, Uzkent B, Tanmay K, et al. Efficient poverty mapping from high resolution remote sensing images[C]//Proceedings of the AAAI Conference on Artificial Intelligence. 2021, 35(1): 12-20.
>
> [2] Xie M, Jean N, Burke M, et al. Transfer learning from deep features for  remote sensing and poverty mapping[C]//Proceedings of the AAAI  conference on artificial intelligence. 2016, 30(1).
>
> [3] Jean N, Burke M, Xie M, et al. Combining satellite imagery and machine  learning to predict poverty[J]. Science, 2016, 353(6301): 790-794.

---

> > ### Comment · Reviewer_phN3 · 2024-08-26
> >
> > Thanks a lot for your response. I have updated my initial score to reflect the concerns addressed by the authors.

---

> > > ### Author Response · Authors · 2024-08-29
> > >
> > > Thank you so much for taking the time to review our revisions. We truly appreciate your thoughtful feedback in improving the quality of our work and your willingness to reconsider our paper.

---

### Official Review · Reviewer_xfSy · 2024-07-25
**Useful contribution for a unified data pipeline**

**Rating:** 6
**Confidence:** 5
**Clarity:** In general good.

**Review:**

The strength are summarized above.

Also after checking the supplementary material, what I miss is that how exactly the data layer can be used for the tasks.

The code description is hidden in the supplementary material (https://github.com/SJTU-CILAB/udl), plz move that to the main text and add to the docs.io (https://urbandatalayer-doc.readthedocs.io/en/latest/index.html) for better overview.

The authors did not provide with coding examples of how to use the layer in concrete tasks (like build-up area prediction) apart from loading the libraries  -- they have mentioned all the sources used for Shanghai and New York, but it would definitely help the community if the processing and visualisation code is also open-sourced. I have a few concrete questions about data sources:
- How are they loaded? Can they be imported from the pip library upon installation?
- but how to extend it with new data sources?
- how are the satellite images used?

Because downloading the data and making them fit into the UDL can be quite challenging without the support from the developers. As we all know from practice, researchers struggle to have a common benchmark because no coding snippets  on tasks are open-sourced.

Hence I will strongly argue that the authors have to provide such a codebase on tasks  -- in general the work quality of the authors is quite high, I am confident that they should be able to provide coding examples. I strongly encourage the AC & SAC to make sure of that.

**Strengths:**

See above in "Review".

**Additional Feedback:**

For me, the paper is 6.5 (between 6-7), i.e., I'd like to accept it conditioned that the authors deliver data set, code availability, task benchmarking wrt other open products like GHSL. I am insisting this because only by doing so the work can have real impact, reproducibility, and usability for the field.

**Correctness:**

In general good.

- ll. 5-6: One might not be able to achieve that in urban planning like CV and NLP due to data privacy and sharing. But this is debatable, I'd say. Hence, I will ask the authors to define the scope of their what, to what extent UDL is useful (data source, tasks, analysis layer, etc.)

**Documentation:**

See points in "Review".

**Ethics:**

The authors can discuss privacy issues in relation to the data sources used / to be collected.

**Limitations:**

What about data availability issues? The UDL cannot be applied to all sources of data (e.g., satellite images, Sentinel and Landsat have various resolutions; some images are proprietary, e.g., worldview, planet), what are the boundaries there?

The authors mention "global" in UDL, how many cities are already in the dataset or how are the data shared/under what license?

**Opportunities For Improvement:**

See above in "Review" and below on data set, code availability, task benchmarking.

**Relation To Prior Work:**

The authors need to benchmark their work against prior global scale data layers like GHSL, https://human-settlement.emergency.copernicus.eu/.

How is UDL better than GHSL, or how does UDL facilitate multi-modal, multi-source analysis in urban studies?

E.g., the authors can do a benchmark study with GHSL on Shanghai and New York or on various tasks.

**Summary And Contributions:**

The paper presents a unified data pipeline for urban studies, including
- four tasks (PM2.5 concentration prediction, build-up areas classification, identification of admin boundaries, El Nino anomaly detection);
- four stages of data wrapping (raw data source, standard data layer, granularity aligned data, fused data),
- three data processing steps (schema transformation, granularity alignment, feature fusion).


The authors have reviewed extensively the existing literature, dataset and benchmark in urban studies (Fig 1, Tab 1) -- this is a very good practice and serves as a general guidance/survey for new comers in the field.

I like the visualization and writing style of the authors very much, clear, atheistically enjoyable to read.

---

> ### Author Rebuttal · Authors · 2024-08-17
>
> > The authors need to benchmark their work against prior global scale data layers like GHSL, https://human-settlement.emergency.copernicus.eu/.
> >
> > How is UDL better than GHSL, or how does UDL facilitate multi-modal, multi-source analysis in urban studies?
> >
> > E.g., the authors can do a benchmark study with GHSL on Shanghai and New York or on various tasks.
>
> **UDL and GHSL serve different purposes.** UDL is a standardized data pipeline rather than a direct urban data release platform. Our aim is to utilize various downloaded data, including GHSL, and convert them into a unified UDL format. This approach addresses the issue of raw data from different sources having diverse formats, which often can not be directly used as model inputs. Additionally, processed data in different research papers may not be exactly comparable due to variations in intermediate processing steps. UDL seeks to standardize this process, making cross-study comparisons more feasible and reliable.

---

> ### Author Rebuttal · Authors · 2024-08-17
>
> Thanks for your insightful review. Here, we have carefully read the reviews and try to address the concerns as follows.
>
> > Also after checking the supplementary material, what I miss is that how exactly the data layer can be used for the tasks.
> >
> > The code description is hidden in the supplementary material (https://github.com/SJTU-CILAB/udl), plz move that to the main text and add to the docs.io (https://urbandatalayer-doc.readthedocs.io/en/latest/index.html) for better overview.
> >
> > The authors did not provide with coding examples of how to use the layer in concrete tasks (like build-up area prediction) apart from loading the libraries -- they have mentioned all the sources used for Shanghai and New York, but it would definitely help the community if the processing and visualisation code is also open-sourced. I have a few concrete questions about data sources:
> >
> > - How are they loaded? Can they be imported from the pip library upon installation?
> >
> > - but how to extend it with new data sources?
> >
> > - how are the satellite images used?
>
> Thank you for the suggestion. Since we cannot change the pdf now, we will put the code description into the main text and the pointer to the docs in the camera-ready version.
>
> We will provide comprehensive sample code for the "Identification of administrative boundaries" and "pm2.5 prediction" cases. These examples will cover the entire process from data loading to methodology and visualization, addressing your questions:
>
> - **Data loading:** The raw data, including tif files and shapefiles, are publicly available urban datasets. UDL processes these into standardized formats for direct use in models. They are not imported directly from the pip library.
>
> - **Extending with new data sources:** UDL defines standard processing and data formats. Any accessible city data can be processed into the five UDL data formats, like our provided samples. For example, in the case of "pm2.5 prediction", the data processing pipeline includes the following steps:
>
>   1. Prepare the original global data in .tif format.
>   2. Segment the global data into regional subsets for better efficiency, using any GIS tools (e.g. ENVI).
>   3. Load the .tif files into UDL layers. The UDL will produce .pickle files.
>   4. Use UDL APIs to align and fuse different data layers if necessary.
>   5. Directly utilize the fused data for machine learning methods, as UDL supports data loaders.
>
>   Users can propose their own methods using the uniform UDL format, ensuring data consistency and result reproducibility. Additionally, researchers can continuously improve the performance of the methods under the same standard.
>
> - **Satellite image usage:** We will provide a detailed explanation in the doc and tif files can be considered as satellite files. Currently many satellite image data are stored in tif format, which our system can handle.
>
>
>
> ---
>
> > What about data availability issues? The UDL cannot be applied to all sources of data (e.g., satellite images, Sentinel and Landsat have various resolutions; some images are proprietary, e.g., worldview, planet), what are the boundaries there?
> >
> > The authors mention "global" in UDL, how many cities are already in the dataset or how are the data shared/under what license?
>
> (1)  **Any accessible urban data can be used with the UDL.** Given that our intention is to standardize the format of urban data to make research uniformly comparable, the boundary of data source lies in whether the data is publicly available or shareable by the authors.
>
> (2)  The data in Table 2 in the Supplementary Material have been already processed and these data are also used in our case. **Except for the El Nino data from UCI repository, all remaining data sources provide data with global coverage.** Especially, for data like population, nightlight and roadnet, they can be used as base layers for various studies. For instance, the OSM dataset we have processed contains 1,964,857,157 nodes and 2,180,447,343 edges on road networks. This covers the whole globe. The use of these data follows both the license of the raw data (such as WorldPop is the Creative Commons Attribution 4.0 International License) and the UDL license (MIT License).
>
>
>
> ---
>
> > One might not be able to achieve that in urban planning like CV and NLP due to data privacy and sharing. But this is debatable, I'd say. Hence, I will ask the authors to define the scope of their what, to what extent UDL is useful (data source, tasks, analysis layer, etc.)
>
> (1)  **For open data sources** like GHSL that do not involve privacy issues, UDL is directly applicable.
>
> (2)  **For sensitive data,** we recommend removing privacy-related information during processing. For example, point data can be converted to grid data, removing metadata for individual points. This process can be done utilizing UDL.
>
> (3)  **In cases where privacy concerns still prevent data sharing, UDL is still especially useful**, because it presents a standard data format to help others replicate experiments under similar conditions. For instance, if built-up surface data is not available for a specific city, researchers can use data from a different city to replicate the experiment, while still using the same UDL grid format to ensure consistency.
>
> (4)  **In addition, users can also use UDL offline to process their own dataset.**

---

> ### Author Response · Authors · 2024-08-29
>
> Thank you very much for your thoughtful feedback and we’ve made several important revisions and clarifications based on your insightful feedback. We sincerely hope that these changes reflect our understanding of your feedback and improve the quality of our work. Your suggestion is very important to us and we would greatly appreciate it if you could review our rebuttal and let us know if the modifications have addressed your concerns before the end of August.

---

### Official Review · Reviewer_xojT · 2024-07-25

**Rating:** 6
**Confidence:** 4

**Review:**

### Pros

- This framework provides a unified framework for data processing. It enhances the reproducibility of results and the ability to benchmark different methodologies, which can accelerate advancements in urban research.
- The paper standardize data pipeline for urban science and computing. This is crucial for enhancing data consistency and comparability across studies, addressing a significant gap in the current fragmented approaches.

### Cons

* The baseline methods in Section 4 rely **primarily on classical machine learning models**. I suggest incorporating more recent and advanced methods to better demonstrate how this unified framework can boost the latest research.
* Promoting standardized methods risks **overlooking the unique contexts** and nuances of different urban environments, which may require tailored approaches instead of a one-size-fits-all solution. For instance, UDL only provides five basic feature fusion methods (“concat,” “sum,” “avg,” “max,” “min”), while more advanced information aggregation methods may be needed for sophisticated applications or research. The authors should consider adding more APIs or allowing users to define their own customized methods.
* While UDL supports multi-modal inputs, the authors **neglect more widely used modalities** in urban science scenarios, such as image, time-series, and text. The authors should consider expand their framework to incorporate these ubiquitous modalities for a more comprehensive data processing pipeline and benchmarking.

**Strengths:**

* UDL provides a unified data framework that addresses the challenges of fragmented data management and inconsistent data formats often encountered in urban research.
* UDL promotes the development of multi-modal urban foundation models by integrating various types of urban data
* By promoting standardization and reproducibility, UDL fosters collaboration within the research community

**Additional Feedback:**

NA

**Clarity:**

* This paper is well-organized and easy to follow.

**Correctness:**

* The paper accurately reflects its contributions and scope. It properly addresses a important limitation in the urban research.

**Documentation:**

* **Pros**: The documentation is complete and well-organized.
* **Cons**: The authors might consider adding toy data samples and quick-start examples to help users understand how to utilize the complete data processing pipeline. This would enhance the usability and accessibility of this tool.

**Limitations:**

* See Cons

**Opportunities For Improvement:**

* See Cons

**Relation To Prior Work:**

* The paper on UrbanDataLayer (UDL) establishes its significance in relation to prior work by addressing the challenges of fragmented data management and processing that have been prevalent in urban research.
* The paper positions UDL as a solution designed to accelerate progress in urban science by fostering collaboration and standardization, ultimately improving the efficiency of the research process.
* Unlike previous papers focusing on data standardization, UDL aims to define a comprehensive urban data pipeline that seamlessly processes and fuses data directly as input into the model. This goes beyond merely providing a platform for data release and access.

**Summary And Contributions:**

This paper addresses the fragmented nature of urban research, which impedes progress due to varied data processing methods and multi-modal data disparities. The authors introduce UDL, a standardized data structure and pipeline for city data engineering. UDL provides a unified data format, enabling researchers to build large-scale benchmarks and integrate multi-modal data, thereby accelerating the development of multi-modal urban foundation models.

---

> ### Author Rebuttal · Authors · 2024-08-17
>
> Thanks for your insightful review. Here, we have carefully read the reviews and try to address the concerns as follows.
>
>
>
> > 1. The baseline methods in Section 4 rely primarily on classical machine learning models.
>
> **The scope of UDL is to simplify and speed up the data processing procedures, rather than proposing any new machine learning models.** In these cases, UDL only affects the processing of the city data (i.e., the inputs of the models) and does not have an impact on the model itself, so UDL does not depend on any model. Hence, we majorly show classical machine learning models just to demonstrate the effectiveness of UDL in converting the data. As an example, in the case of anomaly detection, we further give the results of both the state-of-the-art and classical methods.
>
> ---
>
> > 2. Promoting standardized methods risks overlooking the unique contexts and nuances of different urban environments, which may require tailored approaches instead of a one-size-fits-all solution.
>
> The observation is correct and nuances do exist in the city. Unlike urban data formats that can be fully standardized, feature fusion has been designed to be tailored across different approaches, and we give only the most basic and generic methods.
> **To address this issue, we have added support for function pointers in fusion methods.** This allows users to provide their own customized aggregation methods. These functions must accept an array containing values corresponding small grids and return the aggregated value for the large grid.
> We recognize that user-designed feature fusion methods often have diverse inputs and intermediate processes , making it challenging to provide a uniform definition. In cases where the function pointer definition does not meet the demand, we also welcome extensions and updates to the UDL fusion methods through pull requests.
>
> ---
>
> > 3. While UDL supports multi-modal inputs, the authors neglect more widely used modalities in urban science scenarios, such as image, time-series, and text. The authors should consider expand their framework to incorporate these ubiquitous modalities for a more comprehensive data processing pipeline and benchmarking.
>
> Yes, UDL supports multi-modal inputs, including images, time series, and text, though some additional data preprocessing may be required.
>
> **In urban data science problems discussed in this paper, spatial and temporal information are usually the key. For those data that are completely unrelated with locations, they are out of scope for this paper.** However, for data that do relate to spatial or temporal aspects, we propose the following approaches:
>
> - **Image data:** each image is represented as a vector (containing its location and value), forming a grid or point where values are aggregated from all images in the grid. This image can be geo-tagged images, or raster image. For time-varying images, this creates a multi-layer representation of the grid or point.
> - **Time series data:** processed as point data, with time points serving as attributes.
> - **Text data:** Text with hashtags is treated as a vector containing location and value, and processed in a similar way as image data. As for plain text content, a future direction is to employ Large Language Models (LLMs) to extract grid-wise information from the context.
>
> **It is important to note that associating data with specific locations inherently involves transforming that data into a UDL-compatible layer, such as a grid, graph, point, or polygon.** This is because spatial information must be representable on a map, and UDL encompasses these common types of spatial layers.  We can accommodate user-defined intermediate processing steps to be passed as function pointers, but the final output must still conform to one of the defined UDL layers to ensure consistency. Furthermore, should new multi-modal spatial representations emerge, we are open to expanding the UDL to include additional layer types, thereby enhancing its applicability to a wider range of urban data scenarios.
>
>
> ---
>
>
> > 4. The authors might consider adding toy data samples and quick-start examples to help users understand how to utilize the complete data processing pipeline. This would enhance the usability and accessibility of this tool.
>
> **We appreciate your suggestion, and we will give the complete sample process of “Identification of administrative boundaries” and “pm2.5 prediction” cases as toy samples in the original git repository.** These will include the entire process from data loading and methodology to visualization, helping users better understand UDL. For example, in the case of "pm2.5 prediction", the data processing pipeline includes the following steps:
>
> 1. Prepare the original global data in .tif format.
> 2. Segment the global data into regional subsets for better efficiency, using any GIS tools (e.g. ENVI).
> 3. Load the .tif files into UDL layers. The UDL will produce .pickle files.
> 4. Use UDL APIs to align and fuse different data layers if necessary.
> 5. Directly utilize the fused data for machine learning methods, as UDL supports data loaders.
>
> We are also refining our documentation to further explain the complete data processing pipeline.

---

> > ### Comment · Reviewer_xojT · 2024-09-01
> > **acknowledgement**
> >
> > Thanks for the clarification, I will raise my score to 6.

---

> ### Author Response · Authors · 2024-08-29
>
> Thank you for your valuable feedback on our paper. We have carefully considered your comments and made substantial revisions to address the concerns you raised. We sincerely hope these changes reflect our commitment to improving the quality of our work. We would greatly appreciate it if you could review our rebuttal and let us know if the modifications have addressed your concerns before the end of the rebuttal (the end of August).

---

### Author Response · Authors · 2024-08-22

Dear reviewers and meta-reviewers,

Thank you for your valuable feedback regarding the inclusion of toy examples. We have now updated the code for "Identification of Administrative Boundaries" and "PM2.5 Prediction" cases in the original GitHub repository ([link](https://github.com/SJTU-CILAB/udl)). The data used for these examples can be accessed via [Feishu](https://fi18zswwpe9.feishu.cn/drive/folder/QBujfLaJil09IvdDCYncD2yZnXe?from=from_copylink) or [GoogleDrive](https://drive.google.com/drive/folders/1YuKxh8dIQDnUSgg3VLG_KpPV6K4XKymb?usp=sharing).

Additionally, we are committed to continuously improving the quality of our code and documentation. We are currently enhancing our documentation to further clarify the complete data processing pipeline.

We appreciate your continued feedback and support.

---

### Decision · Program_Chairs · 2024-09-26

**Decision:**

Accept (Poster)

**Comment:**

The framework presented in this paper offers a unified approach to data processing that enhances reproducibility and benchmarking in urban research, addressing significant gaps in current fragmented methodologies. By standardizing the data pipeline for urban science and computing, it improves data consistency and comparability across studies. However, the reliance on classical machine learning models suggests a need for more advanced techniques to showcase the framework's capabilities fully. While UDL supports multi-modal inputs, it overlooks widely used modalities like images, time-series data, and text, limiting its comprehensiveness. The authors should expand the framework to include these modalities and provide more advanced feature fusion methods, as the current options are basic. Additionally, the absence of coding examples for practical tasks, such as area prediction, hampers usability; open-sourcing processing and visualization code would significantly benefit the research community. Questions about data loading, integration of new sources, and satellite image usage remain unaddressed, highlighting a need for clearer guidance. Despite these limitations, the study effectively identifies challenges in urban computing and establishes a robust pipeline for data transformation, granularity alignment, and feature fusion, validated through four case studies. The presentation is clear and logical, making the innovative approach accessible. Overall, while the work is of high quality and has the potential to impact urban science significantly, addressing these points could enhance its effectiveness and usability further.